# Curriculum Disentangled Recommendation with Noisy Multi-feedback

**Hong Chen**[1]*, **Yudong Chen**[1,2]*, **Xin Wang**[1]†,
**Ruobing Xie**[2], **Rui Wang**[2], **Feng Xia**[2], **Wenwu Zhu**[1]†
[1]Tsinghua University, [2]WeChat Search Application Department, Tencent.
{h-chen20,cyd18}@mails.tsinghua.edu.cn, {xin_wang,wwzhu}@tsinghua.edu.cn,
{ruobingxie,rysanwang,xiafengxia}@tencent.com

## Abstract

Learning disentangled representations for user intentions from multi-feedback (i.e., positive and negative feedback) can enhance the accuracy and explainability of recommendation algorithms. However, learning such disentangled representations from multi-feedback data is challenging because i) multi-feedback is complex: there exist complex relations among different types of feedback (e.g., *click, unclick,* and *dislike*, etc) as well as various user intentions, and ii) multi-feedback is noisy: there exists noisy (useless) information both in features and labels, which may deteriorate the recommendation performance. Existing disentangled recommendation works only focus on positive feedback, failing to handle the complex relations and noise hidden in multi-feedback data. To solve this problem, in this work we propose a Curriculum Disentangled Recommendation (CDR) model that is capable of efficiently learning disentangled representations from complex and noisy multi-feedback for better recommendation. Concretely, we design a co-filtering dynamic routing mechanism which simultaneously captures the complex relations among different behavioral feedback and user intentions as well as denoise the representations in the feature level. We then present an adjustable self-evaluating curriculum that is able to evaluate sample difficulties for better model training and conduct denoising in the label level via disregarding useless information. Our extensive experiments on several real-world datasets demonstrate that the proposed CDR model can significantly outperform several state-of-the-art methods in terms of recommendation accuracy[3].

## 1 Introduction

Recommenders aim to capture the user's preferences from different aspects of information for more accurate prediction[1–4] . Learning disentangled representations that can uncover and disentangle the latent explanatory factors hidden in user behavioral data has recently been shown as an effective way to discover users' intentions, improving both recommendation accuracy and explainability [5–9]. Multi-feedback, normally including the positive feedback (e.g., *click*) and negative feedback (*unclick* and *dislike*, etc.), is of great significance to depict the user's unbiased and various intentions [10]. Learning disentangled representation from multi-feedback is able to capture various user intentions more accurately, leading to improvement of accuracy and explainability in recommendation.

However, learning such disentangled representation from multi-feedback that can best serve recommendation is quite challenging due to two reasons. i) **Multi-feedback is complex:** different types of

---

*Equal contributions
†Corresponding Authors.

[3]Our code will be released at `https://github.com/forchchch/CDR`

35th Conference on Neural Information Processing Systems (NeurIPS 2021).

feedback in multi-feedback data have complex relations with each other, and the relations between multi-feedback and various user intentions are also complicated. For instance, a user may *unclick* an item because of disinclination, getting tired of seeing the same type of item previously clicked too many times, as well as lacking enough time to view the item in detail. ii) **Multi-feedback is noisy:** the large amount of feedback such as *unclick* brings a lot of noisy information (useless for recommendation) that may severely deteriorate the recommendation accuracy. For example, a user may *unclick* an item because he truly *dislikes* this item or he has interests in the item but is interrupted by others before making the *click* decision. As such, completely incorporating the unclick feedback as historical features to extract user intention representations may run the risk of bringing **feature-level noise**, while directly regarding the unclick behavior as negative samples to train the model will probably introduce **label-level noise**. Existing works on disentangled representation learning merely rely on the positive user feedback to extract the user intentions, failing to handle the complex relations and noisy (useless) information hidden in multi-feedback data.

To tackle these challenges, we propose a Curriculum Disentangled Recommendation (CDR) model that is able to accurately discover various user intentions from different kinds of user feedback. The CDR model consists of two core components, i.e., the co-filtering dynamic routing mechanism and the adjustable self-evaluating curriculum strategy, which together learn disentangled representations as well as captures the complex relational dependencies and filters out noise from multi-feedback to achieve more accurate recommendation. More concretely, the proposed routing mechanism utilizes the dependencies among different feedback to accurately discover the user preferences, followed by the intention aggregation which further helps to learn disentangled representations for users. Our adjustable self-evaluating curriculum then guides the model towards better optima by reweighing samples according to the self-evaluated difficulty. The proposed curriculum can further enhance the model performance by making the model learn from multi-feedback data of different difficulties at different learning paces, thus preventing the model from overfitting the data with improper labels. We conduct extensive experiments on several real-world datasets to demonstrate that our proposed CDR model can significantly beat baseline methods in terms of recommendation accuracy.

To summarize, our contributions are listed as follows. (1) We propose a Curriculum Disentangled Recommendation (CDR) model to learn disentangled representations for recommendation from multi-feedback with complex relations and noisy information. (2) We propose a co-filtering dynamic routing mechanism to simultaneously capture complex relations among different sorts of feedback and various user intentions as well as filtering out feature-level noise in multi-feedback. (3) We propose an adjustable self-evaluating curriculum to guide the model towards better optima by alleviating the impact of label-level noise in a more controllable and convenient way.

## 2 Related Work

**Recommendation Based on Multi-Feedback** Apart from positive feedback, negative feedback data are also essential for capturing user preference [11]. Early efforts [12–14] regard all unclick data equally as negative feedback and decrease the confidence compared to click data. Later works utilize additional information [15–18] or reinforcement learning [19–21] to distinguish real negative signals from all the unclick data. A recent method [22] also recognizes negative feedback from the clicked news by reading dwell time. However, all these methods merely select better negative training samples and ignore multi-feedback as features to learn user interests. DFN [10] simultaneously incorporate click, unclick, and dislike feedback simply by rough attention mechanism and concatenation, which is insufficient to capture the user's comprehensive and accurate interests.

**Disentangled Representation Learning** Disentangled representation learning aims to learn various hidden explanatory factors behind observable data in different parts of the learned vector presentation [23]. Many variants of variational auto-encoder (VAE) [24] have been studied to improve the disentanglement of the learned representation [25–27] by adding regularization terms to decrease the mutual information between different parts of the learned vector. Disentangled representation learning has also found its application in recommendation [5–8, 28, 29] by learning disentangled user preference from user positive feedback to improve both the performance and interpretability. Different from these works, our work focuses on learning more comprehensive disentangled interests with multi-feedback.

**Curriculum Learning** Curriculum learning [30, 31] aims to design a dynamic sample reweighting strategy throughout the training process to improve performance and training efficiency. Traditional methods mostly follow an easy-to-hard paradigm, i.e., assigning higher weights to easier samples in earlier training, where the "easiness" measurement can be both domain-knowledge-based [32–34] and loss-based [35, 36]. These methods can only be effective for specific tasks, and "easy-to-hard" is sometimes sub-optimal compared with "harder first" [37]. Recent works propose to automatically learn a curriculum by reinforcement learning [38–40], meta-learning [41], etc. However, the optimization process of these curricula is time- or resource-consuming, which is too costly for abundant recommendation data. Our method provides an efficient and also flexible curriculum that is beyond the easy-to-hard limit and more adjustable and interpretable than automatic methods.

## 3 Method

In this section, we introduce the proposed Curriculum Disentangled Recommendation (CDR) model (Figure 1) to mine comprehensive and accurate user intentions from users' noisy multi-feedback.

### 3.1 Notations and Problem Formulation

**Notations** We denote $\phi(\mathbf{a}, \mathbf{b})$ as the inner product of two vectors $LayerNorm(\mathbf{a})$ and $LayerNorm(\mathbf{b})$, where $LayerNorm$ refers to Layer Normalization operation and $\phi(\mathbf{a}, \mathbf{b})$ measures the similarity between vector $\mathbf{a}$ and $\mathbf{b}$. $sim(\mathbf{key}_i, \mathbf{query})$ is the normalized similarity between $\mathbf{query}$ and $\mathbf{key}_i$ on set $Q$ that is composed of all the keys: $sim(\mathbf{key}_i, \mathbf{query}) = \frac{\exp(\phi(\mathbf{key}_i, \mathbf{query}))}{\sum_{i' \in Q} \exp(\phi(\mathbf{key}_{i'}, \mathbf{query}))}$.

**Multi-feedback based prediction** The $v^{th}$ user's historical behavior contains his or her **clicked** item sequence $c^{(v)} = [c_1^{(v)}, c_2^{(v)}, \cdots, c_m^{(v)}]$, **unclicked** (i.e., presented but not clicked) item sequence $u^{(v)} = [u_1^{(v)}, u_2^{(v)}, \cdots, u_n^{(v)}]$, and **disliked** (i.e., press to the dislike button or low rated) item sequence $d^{(v)} = [d_1^{(v)}, d_2^{(v)}, \cdots, d_l^{(v)}]$, where each term in these sequences represents an item that the $v^{(th)}$ user interacted with. Our goal is to learn users' disentangled intentions and then accurately predict $s_{vt}$, i.e., the $v^{th}$ user's preference towards the candidate item $t^{(v)}$. Besides, we utilize the profile (e.g., age and gender) of the $v^{th}$ user to aid the learning and prediction process.

### 3.2 Co-filtering Dynamic Routing

The routing mechanism takes as input user profile, candidate item feature, and user multi-feedback history, and makes the final prediction by the following three steps. It first utilizes the relations behind different kinds of feedback to discover where the user's true interests locate. Then, the model aggregates the user's disentangled intentions from the useful behavior. Finally, it predicts the user's preference towards the candidate item based on the learned intentions.

#### 3.2.1 Interests Mining

The user's interests could be reflected by his or her various kinds of feedback. Ignoring any kind of feedback may lead to incomplete or inaccurate preference modeling. However, the noise hidden behind these data makes it infeasible to directly use multi-feedback to learn the user's interests. Thus, how to make use of the relations of different kinds of feedback to discover where the user's true interests locate is the key in this step.

First, we project each item in $c^{(v)}$, $d^{(v)}$, and $u^{(v)}$ into the embedding space by concatenating its ID and category embedding. Thus, we obtain the clicked embedding sequence $\mathbf{h}_c^{(v)} = [\mathbf{h}_{c1}^{(v)}, \mathbf{h}_{c2}^{(v)}, \cdots, \mathbf{h}_{cm}^{(v)}]$, unclicked embedding sequence $\mathbf{h}_u^{(v)} = [\mathbf{h}_{u1}^{(v)}, \mathbf{h}_{u2}^{(v)}, \cdots, \mathbf{h}_{un}^{(v)}]$, and disliked embedding sequence $\mathbf{h}_d^{(v)} = [\mathbf{h}_{d1}^{(v)}, \mathbf{h}_{d2}^{(v)}, \cdots, \mathbf{h}_{dl}^{(v)}]$. We also project the information of the user profile into the embedding space and obtain the user profile feature $\mathbf{F}^{(v)} = \{\mathbf{F}_1^{(v)}, \mathbf{F}_2^{(v)}, \cdots, \mathbf{F}_g^{(v)}\}$, where $g$ is the number of user profiles. We then utilize the power of transformer encoder [42] to obtain better representation for each item with fully interactions: $[\mathbf{z}_{c1}^{(v)}, \mathbf{z}_{c2}^{(v)}, \cdots, \mathbf{z}_{cm}^{(v)}] = C\text{-}Encoder(\mathbf{h}_c^{(v)})$, $[\mathbf{z}_{u1}^{(v)}, \mathbf{z}_{u2}^{(v)}, \cdots, \mathbf{z}_{un}^{(v)}] = U\text{-}Encoder(\mathbf{h}_u^{(v)})$, $[\mathbf{z}_{d1}^{(v)}, \mathbf{z}_{d2}^{(v)}, \cdots, \mathbf{z}_{dl}^{(v)}] = D\text{-}Encoder(\mathbf{h}_d^{(v)})$, where the $\mathbf{z}_{ci}^{(v)}, \mathbf{z}_{ui}^{(v)}, \mathbf{z}_{di}^{(v)}$ refer to $d$-dimensional feature vectors containing the $v^{th}$ user's interests.

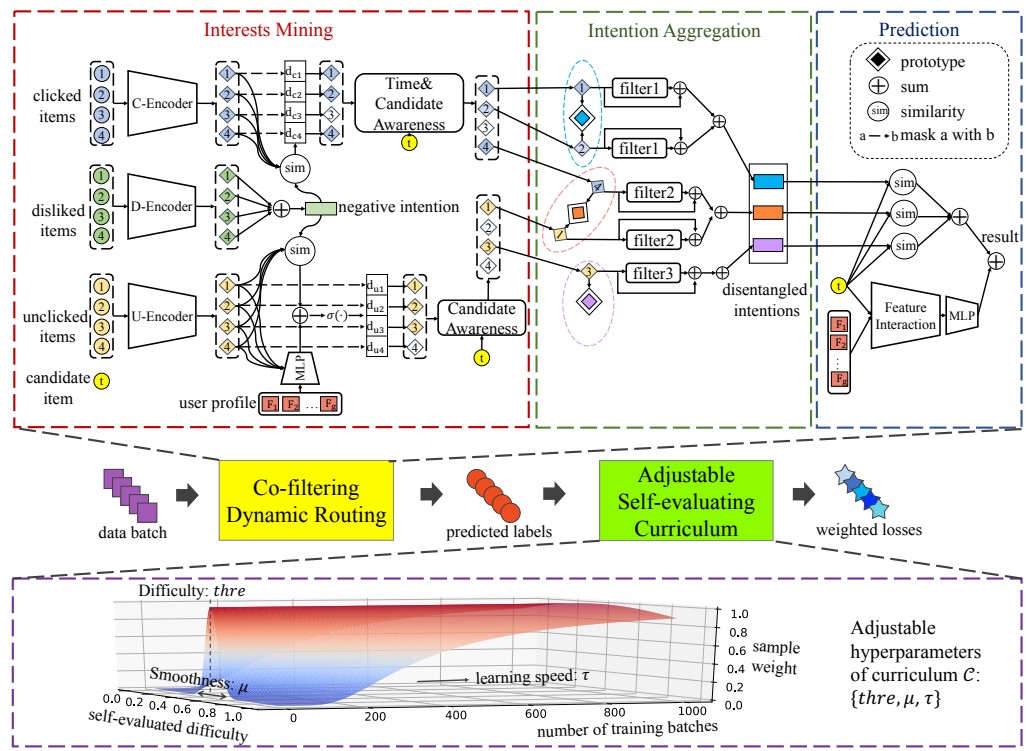

Figure 1: The framework of the proposed Curriculum Disentangled Recommendation (CDR) model

We then need deeper thinking towards the relations among multiple feedback to discover user's true interests. The first thing to consider is where the user's interests will locate. We note that not only the click behavior contains user interests, but the unclick sequence generated by the recommender algorithms could also contain rich information about user interests. Therefore, utilizing both kinds of feedback could help to learn comprehensive user interests. The second noteworthy thing is how to discover the user's true interests from the noisy data. To filter out the noise information in these historical sequences, we need the disliked item sequences that could reflect the user's strong negative intentions and have high confidence. Thus, we average these strong negative features and regard it as the negative intention of the $v^{th}$ user: $\mathbf{n}^{(v)} = \frac{1}{l}\sum_{i=1}^{l}\mathbf{z}_{di}^{(v)}$. This highly confident negative intention could be used to filter the noise in both clicked and unclicked history items. We adopt the similarity between each item and the negative intention to judge whether the item should contribute to the user intentions. Based on the intuition that higher similarity to the negative intention should contribute less to the user preference, the contribution of each item to the user intentions can be formulated as follows:

$$
\begin{aligned}
&\mathbf{n}_{c}^{'(v)} = \mathbf{n}^{(v)} + \mathbf{b}_{c1}, \ \mathbf{z}_{ci}^{'(v)} = \mathbf{z}_{ci}^{(v)} + \sigma_1(\mathbf{W}_{c1}\mathbf{z}_{ci}^{(v)} + \mathbf{b}_{c2}), \\
&d_{ci}^{(v)} = \frac{exp(-\phi(\mathbf{z}_{ci}^{'(v)}, \mathbf{n}_{c}^{'(v)}))}{\sum_{j=1}^{m} exp(-\phi(\mathbf{z}_{cj}^{'(v)}, \mathbf{n}_{c}^{'(v)}))}, \\
&\mathbf{n}_{u}^{'(v)} = \mathbf{n}^{(v)} + \mathbf{b}_{u1}, \ \mathbf{z}_{uj}^{'(v)} = \mathbf{z}_{uj}^{(v)} + \sigma_1(\mathbf{W}_{u1}\mathbf{z}_{uj}^{(v)} + \mathbf{b}_{u2}), \\
&d_{uj}^{(v)} = \sigma(-\phi(\mathbf{z}_{uj}^{'(v)}, \mathbf{n}_{u}^{'(v)}) + MLP([\mathbf{F}^{(v)}; \mathbf{z}_{uj}^{(v)}])),
\end{aligned}
\tag{1}
$$

where $\sigma_1$ is the ReLU activation function and $\sigma$ is the Sigmoid activation function. $d_{ci}^{(v)}$ and $d_{uj}^{(v)}$ represent the contribution of each clicked item and unclicked item to the user intention, respectively. We obtain $d_{ci}^{(v)}$ by calculating the similarity between each clicked item and the negative intention, and we also introduce the parameters $\mathbf{W}_{c1}, \mathbf{b}_{c1}, \mathbf{b}_{c2}$ to avoid some outliers that disobey our intuition. As for the unclicked item sequence, there exists more noise. Thus, we utilize not only the information of the user negative intention but also the user profile feature $\mathbf{F}^{(v)}$ to measure the importance of each

unclicked item. We first concatenate user profile features and the unclicked item feature together, and then use a two-layer MLP to judge whether the unclicked item is important.

Besides the noise effect, we also take the time and candidate item factors into consideration. Intuitively, more recently clicked items and items having a higher similarity to the candidate item will contribute more to the user preference towards the candidate item. The formulation is as follows:

$$
\begin{aligned}
& \textbf{time}_{ci}^{(v)} = \mathbf{z}_{ci}^{(v)} + \sigma_1(\mathbf{W}_{c2}[\mathbf{z}_{ci}^{(v)}; \mathbf{p}_i] + \mathbf{b}_{c3}), \ \textbf{time}_{cm}^{'(v)} = \mathbf{z}_{cm}^{(v)} + \sigma_1(\mathbf{W}_{c3}[\mathbf{z}_{cm}^{(v)}; \mathbf{p}_m] + \mathbf{b}_{c4}), \\
& \textbf{cand}_{ci}^{(v)} = \mathbf{z}_{ci}^{(v)} + \sigma_1(\mathbf{W}_{c4}\mathbf{z}_{ci}^{(v)} + \mathbf{b}_{c5}), \ \mathbf{z}_{ct}^{'(v)} = \mathbf{h}_t^{(v)} + \mathbf{b}_{c6}, \\
& f_{ci}^{(v)} = sim(\textbf{time}_{ci}^{(v)}, \textbf{time}_{cm}^{'(v)}) + sim(\textbf{cand}_{ci}^{(v)}, \mathbf{z}_{ct}^{'(v)}), \\
& \textbf{cand}_{uj}^{(v)} = \mathbf{z}_{uj}^{(v)} + \sigma_1(\mathbf{W}_{u2}\mathbf{z}_{uj}^{(v)} + \mathbf{b}_{u3}), \ \mathbf{z}_{ut}^{'(v)} = \mathbf{h}_t^{(v)} + \mathbf{b}_{u4}, \ f_{uj}^{(v)} = sim(\textbf{cand}_{uj}^{(v)}, \mathbf{z}_{ut}^{'(v)}).
\end{aligned}
\tag{2}
$$

We consider the impact of both time and candidate factors for the clicked items. The candidate item feature $\mathbf{h}_t^{(v)} \in \mathbb{R}^d$ and the most recently clicked item feature $\mathbf{z}_{cm}^{(v)}$ measures the importance of each clicked item. $\mathbf{p}_i$ is the position embedding of each item and encoded for more accurate time factor consideration. However, for the unclick sequence, we only consider the candidate factor, because the time of unclicking one item has low relations with the user's current interests. $f_{ci}^{(v)}$ and $f_{uj}^{(v)}$ are the importance of each clicked item and unclicked item, respectively.

### 3.2.2 Intention Aggregation

After considering how each kind of feedback will contribute to the user's intentions, we then conduct intention aggregation to obtain the $v^{th}$ user intentions under various latent categories. Assuming that the intentions of all users can be decomposed to $K$ latent categories and each latent category has its prototype $\mathbf{m}_k \in \mathbb{R}^d$, $k = 1, 2, \cdots, K$, we can predict the probability of each item belonging to the $k^{th}$ latent category by their similarities:

$$
c_{ik}^{(v)} = sim(\mathbf{m}_k, \mathbf{z}_{ci}^{(v)}), \ u_{jk}^{(v)} = sim(\mathbf{m}_k, \mathbf{z}_{uj}^{(v)}).
\tag{3}
$$

We then use the highly confident negative intention to filter the click and unclick historical item features by a residual structure. The $k^{th}$ intention of the $v^{th}$ user could be aggregated from all the filtered clicked and unclicked item features, $\mathbf{z1}_{ci}^{(v)}$ and $\mathbf{z1}_{uj}^{(v)}$ as Eq. (4). $\boldsymbol{\beta}_k$ is the bias for the $k^{th}$ latent category. $\lambda < 1$ is the prior confidence of the unclicked item. $\textbf{inten}_k^{(v)}$ is the $v^{th}$ user's interest under the $k^{th}$ latent category.

$$
\begin{aligned}
& \mathbf{z1}_{ci}^{(v)} = \mathbf{z}_{ci}^{(v)} + MLP([\mathbf{z}_{ci}^{(v)}; \mathbf{n}_c^{'(v)}; \mathbf{z}_{ci}^{(v)} - \mathbf{n}_c^{'(v)}; \mathbf{z}_{ci}^{(v)} \odot \mathbf{n}_c^{'(v)}]), \\
& \mathbf{z1}_{uj}^{(v)} = \mathbf{z}_{uj}^{(v)} + MLP([\mathbf{z}_{uj}^{(v)}; \mathbf{n}_u^{'(v)}; \mathbf{z}_{uj}^{(v)} - \mathbf{n}_u^{'(v)}; \mathbf{z}_{uj}^{(v)} \odot \mathbf{n}_u^{'(v)}]), \\
& \textbf{inten}_k^{(v)} = \sum_{i=1}^m d_{ci}^{(v)} \cdot f_{ci}^{(v)} \cdot c_{ik}^{(v)}(\mathbf{z1}_{ci}^{(v)} + \boldsymbol{\beta}_k) + \lambda \sum_{j=1}^l d_{uj}^{(v)} \cdot f_{uj}^{(v)} \cdot u_{jk}^{(v)} \cdot (\mathbf{z1}_{uj}^{(v)} + \boldsymbol{\beta}_k).
\end{aligned}
\tag{4}
$$

### 3.2.3 Prediction

Based on the learned intentions, we can predict the user's preference towards the candidate item by calculating the inner product between the candidate item and user disentangled intentions. However, there also exist some users who might have little historical behaviors. So we also manage to discover the interests of the $v^{th}$ user from his or her profile feature and the candidate item feature, aiming to capture some common patterns, e.g., females may show high interests in fashion articles. We collect these information in $\boldsymbol{F_{all}} = [\mathbf{F}_1^{(v)}, \cdots, \mathbf{F}_g^{(v)}, \mathbf{h}_{t1}^{(v)}, \cdots, \mathbf{h}_{tr}^{(v)}]$, where $\mathbf{h}_{tj}^{(v)} \in \mathbb{R}^{\frac{d}{r}}$ is one field of the candidate item embedding $\mathbf{h}_t^{(v)}$. Then the multi-head attention [42] is used to capture the interactions of different fields of features:

$$
[\boldsymbol{f}_1, \boldsymbol{f}_2, \cdots, \boldsymbol{f}_{g+r}] = MultiheadAttention(\boldsymbol{F_{all}}).
\tag{5}
$$

By simultaneously considering the user's interests learned from the historical multi-feedback and feature interaction, we can predict the user preference towards the candidate item $s_{vt}$ as Eq. (6), where the first term is the preference inferred from user feedback ($<>$ is the inner product) and the second term is from the feature interaction between the user profile and item features.

$$
s_{vt} = \sum_{k=1}^K < \textbf{inten}_k^{(v)}, h_t^{(v)} > + MLP([\boldsymbol{f}_1; \boldsymbol{f}_2; \cdots; \boldsymbol{f}_{g+r}]).
\tag{6}
$$

Our loss function is designed as follows, composing of the widely adopted binary cross-entropy loss and the regularization term for disentanglement aiming to minimize the similarities among different intentions. $y_{vt}$ is the ground truth label, and $D_T$ is the training set.

$$L = -\frac{1}{|D_T|} \sum_{(v,t) \in D_T} y_{vt} \log \sigma(s_{vt}) + (1 - y_{vt}) \log(1 - \sigma(s_{vt})) + \sum_{i=1}^{K} \sum_{j \neq i} \phi(\mathbf{m}_i, \mathbf{m}_j) \qquad (7)$$

### 3.3 Adjustable Self-evaluating Curriculum towards A Better Self

All the aforementioned methods could help to capture the user's comprehensive true interests from the input features. However, the noise existing in the training labels will also mislead our model to suboptimal parameters. To tackle the problem, we leverage the idea of curriculum learning to denoise [31]. However, it is proved in [36] that different curriculum strategies (e.g., easy to hard, hard example mining, etc.) can be effective for different dataset settings, and thus we need a flexible curriculum design to adapt to complex recommendation scenarios.

To this end, we propose a novel adjustable self-evaluating curriculum for recommendation, which algorithm is shown in Algorithm 1. Our goal is to obtain the optimal parameters $\theta$ for our recommender $F$. When training on the batch $B$, we first calculate the prediction result $\sigma(s_j)$ for each sample and the difference between the ground truth label $y_j$ and our prediction. Then, we obtain the importance of each sample through a Gaussian distribution, where the sample with a closer difference to a preset value $thre$ will get higher importance. Here, the hyperparameter $thre \in (0, 1)$ reflects the sample difficulty we hope the model to focus on. Concretely, if $thre$ approaches 1, the harder samples will get higher weights. While if $thre$ approaches 0, the model will lay more emphasis on the easier samples. In our experiment, we can adjust the values for $thre$ to set the curriculum of different difficulty levels for our model to improve itself. Finally, we optimize the parameters by the reweighted loss with an existing optimization method $\pi$. In the algorithm, $\mu$ controls the degree of concentration of the model and $\tau$ is a time-weight-decay factor. As the training goes on, $\mu$ is becoming smaller and smaller by timing $\tau$, which makes the Gaussian distribution smoother and smoother so that all the samples almost get equal weights for training in the end. This time-varying process conforms to the human learning process: after we gradually improve ourselves by a scheduled curriculum, we should review all the samples to further consolidate the knowledge in our mind. Since in the early stage the model has already learned enough knowledge, it could be more robust and less likely to be influenced by the noisy samples. By adjusting the values of $\tau$, we can control the curriculum learning speed, where a larger $\tau$ means a slower learning process.

---

**Algorithm 1** Adjustable Self-evaluating Curriculum towards A Better Self

---
1: **input:** $\{(x_{vt}, y_{vt})\}_{(v,t) \in D_T}, \pi(\cdot), \ell(\cdot, \cdot), \tau < 1, counter, interval, thre, F$
2: **initialize:** $\theta, \mu$
3: **for** $B \subset D_T$ **do**
4:     **for** $(x_j, y_j) \in B$ **do**
5:       $s_j = F(x_j), d_j = abs(y_j - \sigma(s_j));$
6:       $w_j = exp(-\mu \times (d_j - thre)^2);$
7:       $w_j \leftarrow w_j / \sum_{j=1}^{|B|}(w_j);$
8:     **end for**
9:     $\theta \leftarrow \theta - \pi(\nabla_\theta \sum_{j=1}^{|B|} w_j \ell(s_j, y_j) + \sum_{i=1}^{K} \sum_{j \neq i} \phi(\mathbf{m}_i, \mathbf{m}_j));$
10:    $counter \leftarrow counter + 1;$
11:    **if** $counter\%interval == 0$ **then**
12:       $\mu \leftarrow \mu \times \tau;$
13:    **end if**
14: **end for**

---

**Discussion**. Our proposed curriculum elegantly converts the designing process of a curriculum training strategy to a hyper-parameter search process on $thre$, $\mu$ and $\tau$, which improves the flexibility, controllability, and explainability of curriculum design, while goes beyond the conventional assumption of "easy-to-hard" [30, 35, 32, 33]. Meanwhile, compared to automatic curricula [38, 40, 43, 39], our method requires almost no extra time and memory overhead for learning and applying the curriculum, which is efficient and conforms to the demand of recommendation scenarios.

# 4 Experiments

## 4.1 Experimental Setup

Table 1: Dataset statistics

|            | Amazon-Beauty | Amazon-Sports | MovieLens-1M | WeChat5D |
|------------|---------------|---------------|--------------|----------|
| # of users | 22,342 | 35,590 | 6,039 | 13,340 |
| # of items | 12,099 | 18,356 | 3,628 | 112,859 |
| # of click | 176,520 | 277,088 | 836,478 | 749,138 |
| # of unclick | 788,008 | 1,179,266 | 2,138,040 | 7,766,013 |
| # of dislike | 21,847 | 19,203 | 163,515 | 295,504 |

**Datasets** We conduct our experiments on four real-world datasets: WeChat5D, MovieLens-1M[44], Amazon Sports[45] and Amazon Beauty[45]. The datasets statistics are shown in Table 1. WeChat5D is a mobile article recommendation dataset of WeChat Top Stories and itself has different kinds of user feedback. All data are preprocessed via data masking to protect user privacy. For the Amazon and MovieLens dataset, we regard the ratings that are larger than 2 points as click feedback, while the rest as the dislike feedback. Since these three datasets have no information about the items that are recommended to the users but are unclicked, we simulate a simple recommendation environment. For each piece of user like or dislike interaction, we generate four pieces of unclick interactions for this user. They include three items that are sampled from the top popular items at that time and one item randomly sampled from all the items. This simulates the simplest recommendation rule, i.e., recommendation based on item popularity. The randomly sampled item simulates the scenario that the recommenders would always recommend something to explore the customer's potential interests [46]. Note that all the user's unclick interactions are not in the user clicked or disliked item sets. More specifically, based on the two Amazon datasets and the MovieLens-1M dataset, for each piece of user like or dislike interaction, we generate four pieces of unclick interactions for this user. They include three items that are sampled from the top popular (top 3000 for Amazon datasets and top 1/2 for MovieLens-1M) items at that time and one item randomly sampled from all the items. The whole dataset is chronologically divided to the train, valid, and test dataset by the ratio of 8:1:1. Note our training and testing phase follow the sequential recommendation setting. For example, if one user's historical behavior is a sequence $\{1, 2, 3, \cdots, 18, 19, 20\}$. Then, we will generate the validation and test samples as follows: two validation samples $\{[1, 2, 3, 4, \cdots, 16], [17]\}, \{[1, 2, 3, 4, \cdots, 16, 17], [18]\}$ and two test samples $\{[1, 2, 3, 4, \cdots, 16, 17, 18], [19]\}, \{[1, 2, 3, 4, \cdots, 16, 17, 18, 19], [20]\}$, where the first term in [ ] represents the historical information we use for prediction and the second term is the next item for prediction. We always use all the user's real historical behaviors as the sequential input to the models to predict the next item the user will click.

**Baselines** We compare our approach with the state-of-the-art (SOTA) methods. DeepFM [47] and AutoInt [48] are recommenders based on feature interaction. DIN [49], SASRec [50], DFN [10], SDR [9] are methods based on the user's sequential historical behavior. Specifically, DFN considers different kinds of user feedback and concatenates these features together. SDR only utilizes positive click feedback to capture the user's disentangled interests. For fair comparison, we add the feature interaction module to all the baselines if the model originally does not utilize the user profile.

**Implementation and hyper-parameters** We implement our method in Tensorflow and use the Adagrad [51] optimizer for mini-batch gradient descent that is suitable for sparse data, while the size of each mini-batch is 256. All the mentioned transformer encoders are four-head and one-layer. We cap the maximum sequential historical behavior length to 30 for all datasets. We fix $\mu$ in the curriculum to 10 and the other hyper-parameters are then tuned using random search. The search space is listed as follows. More detailed experimental settings can be found in our appendix.

- The number of latent intentions $K \in \{1, 2, \cdots, 8\}$.

- The prior confidence for the unclicked data $\lambda \in \{0.1, 0.2, \cdots, 1.0\}$.

- The learning rate $\in \{0.0001, 0.001, 0.01, 0.1, 1.0\}$.

- The hidden size of each field of feature $\in \{32, 64, 128, 256\}$

Table 2: Model performance

| Model | Amazon-Beauty | | Amazon-Sports | | MovieLens-1M | | WeChat5D | |
|---|---|---|---|---|---|---|---|---|
| | AUC | RelaImpr | AUC | RelaImpr | AUC | RelaImpr | AUC | RelaImpr |
| DeepFM | 0.6975 | 0.00% | 0.7608 | 0.00% | 0.8098 | 0.00% | 0.7219 | 0.00% |
| AutoInt | 0.6826 | -7.57% | 0.7575 | -1.27% | 0.7940 | -5.08% | 0.7263 | 1.96% |
| SASRec | 0.7415 | 22.28% | 0.7830 | 8.52% | 0.8826 | 19.32% | 0.7126 | -4.18% |
| DIN | 0.7633 | 33.32% | 0.7968 | 13.80% | 0.8760 | 21.39% | 0.7282 | 2.83% |
| DFN | 0.7670 | 35.21% | 0.7763 | 5.93% | 0.8293 | 6.31% | 0.7329 | 4.96% |
| SDR | 0.7238 | 13.30% | 0.7972 | 13.95% | 0.8865 | 24.78% | 0.7296 | 3.45% |
| CDR (Ours) | **0.7991** | **51.43%** | **0.8152** | **20.85%** | **0.9065** | **31.24%** | **0.7622** | **18.14%** |

## 4.2 Recommendation Performance

We evaluate the performance of our proposed method on the classical click-through-rate (CTR) prediction task and utilize a widely-used metric Area Under Curve (AUC) for evaluation. We also follow [52] to use the RelaImpr to measure the relative improvements over the base model (i.e.,DeepFM in our experiment). The results are shown in Table 2.

We observe that our approach outperforms the baselines significantly, both on the dense MovieLens dataset and the sparse Amazon and WeChat5D dataset. We can see that on the MovieLens dataset where each user has rich historical behaviors, the models that utilize the sequential historical behavior obtain much better performance. However, DFN fails to accurately capture the user's interests compared to other sequential models in MovieLens-1M. This is likely because DFN uses three kinds of feedback and in this dataset, the user has rich feedback that contains more noise (introduced by more unclick behavior), while DFN fails to filter the noise. Our method performs best on datasets that have different sparsity, mainly benefiting from both our model design and the curriculum training strategy, which is discussed in Section 4.3 and Section 4.4, respectively.

## 4.3 Multi-feedback

We validate the effectiveness of our method on capturing the user's preference from different kinds of feedback. We compare the following four situations, our complete method (complete), our model without curriculum (w/o CL), our model that only utilizes the clicked historical feedback without curriculum (click), and our model that directly utilizes the clicked and unclicked historical feedback(i,e, the output of the C-Encoder and U-Encoder) to aggregate the user's intentions without curriculum (c&un). The result is shown in Table 3.

Table 3: Effectiveness of our model components

| Model | Amazon-Beauty | | Amazon-Sports | | MovieLens-1M | | WeChat5D | |
|---|---|---|---|---|---|---|---|---|
| | AUC | RelaImpr | AUC | RelaImpr | AUC | RelaImpr | AUC | RelaImpr |
| w/o CL | 0.7914 | 47.56% | 0.8083 | 18.22% | 0.8972 | 28.22% | 0.7548 | 14.82% |
| click | 0.7477 | 25.43% | 0.8006 | 15.26% | 0.8879 | 25.22% | 0.7347 | 5.76% |
| c&un | 0.7477 | 25.42% | 0.7969 | 13.86% | 0.8854 | 24.40% | 0.7340 | 5.42% |
| complete | 0.7991 | 51.43% | 0.8152 | 20.85% | 0.9065 | 31.24% | 0.7622 | 18.14% |

By comparing the click and c&un results, we conclude that directly utilizing the unclick historical behavior will not improve the model performance. A possible reason is that the noise in unclick feedback will prevent the model from capturing the user's true interests. However, with our proposed co-filtering mechanism to filter the noise and locate the user's true preference, the model could more precisely capture the user's intentions and bring higher performance (comparing the results of w/o CL with that of c&un). Furthermore, the results of complete and w/o CL indicate that the curriculum learning strategy takes a further step to help the model to learn better parameters.

## 4.4 Curriculum Exploration

In this part, we explore what kind of curriculum strategy would benefit our model. We respectively explore the impact of the curriculum difficulty $thre$ and the curriculum learning speed $\tau$.

**Impact of Curriculum Difficulty**   As the difficulty of our curriculum is controlled by the parameter $thre$ (larger $thre$ means curriculum of higher difficulty), we set different values for $thre$ to train our model, while fixing the other hyper-parameters the same as the random grid search result. The performance of the model trained with different values of $thre$ is shown in Fig 2(a). The results on Amazon Sports dataset show the typical "easy-to-hard" curriculum pattern [31]. Learning from the easier samples first will make the objective smoother, thus more easily reaching the global optimal. In contrast, the results on the Amazon Beauty dataset show a different curriculum scenario: although learning easy samples first could achieve comparably good performance, learning from the samples that have difficulty of about 0.8 could lead to better performance. It matches the spirits of "hard example mining" [37], focusing more on the harder samples that are more informative for the model helps the model to discriminate the samples better. Although the results on the two datasets show that they have different suitable curriculum, there is one phenomenon in common, curriculum concentrating on the too hard samples ($thre = 1.0$) results in bad performance. This is quite reasonable for our scenario, because there exists label-level noise. While the noisy data always cause higher prediction error than the clean data [53], focusing on the noisy data will absolutely cause the learned parameters sub-optimal.

**Impact of Curriculum Speed**   Fixing the most suitable curriculum difficulty $thre$, we change the hyper-parameter $\tau$ to see at what pace the model should learn. Smaller $\tau$ means faster adapting to all the samples considering that smaller $\tau$ will make the Gaussian function smooth faster. From the results in Fig 2(b), we can see that too fast or too slow curriculum is not good enough, and the best paces for different datasets are different. Moreover, we observe that if $\tau$ equals 1, the performance on both datasets will drop[4]. This phenomenon is easy to understand from human learning. If we always concentrate on the problems of some particular level of difficulty, we cannot perform well in exams when the problems of different levels of difficulty are present to us. For the Amazon Sports dataset, since we fix $thre = 0.0$, it always concentrates on the easiest samples and hardly learns the more difficult samples, and thus suffer dramatically performance drop when test. While for the Amazon Beauty dataset, since $thre = 0.8$, it more focuses on its errors during all the training process, and thus not suffers as much performance drop as the Amazon Sports dataset.

## 4.5   Other Studies on Explainability

**Disentangled Intentions**   We validate the disentanglement of our learned different intentions. We calculate the similarity between each item and each of $K$ prototype intention, and assign each item to the intention that has the highest similarity with it. Then under each intention, we calculate the number of items belonging to each item category and plot the results in Figure 4. We can see that the learned intentions have disentangled meanings. For example, in Figure 4(d), intention 0 means interests in "Sports", while intention 5 mainly means preference towards "Entertainment". However, the disentanglement on the MovienLens-1M dataset is not so promising, almost all intentions are highly related to "Comedy", "Action", and "drama". This phenomenon is probably because the long historical behavior in MovieLens-1M brings great challenges for disentanglement as mentioned in [9].

**Mining Interests from Unclick Sequence**   We also conduct an ablation study to validate our claim that there also exist user's interests in the unclick sequence and our model can locate these interests from these noisy data. One example is shown in Figure 3 and we can see that this user clicked topics about "Social livelihood", "Technology" and "Human history". Our interests mining algorithm discovers the user's interests in "International News" and "Education" from the unclick feedback. When the candidate item about "International News" comes, our model can make the right prediction based on these located interests, showing that our model could mine the user's true interests that do not hide in the clicked history from the user's unclick history, thus capturing users' comprehensive and accurate intentions.

---

[4]we don't plot the performance on Amazon Sports dataset because its performance drops to 0.5645 when $\tau$ equals 1.0.

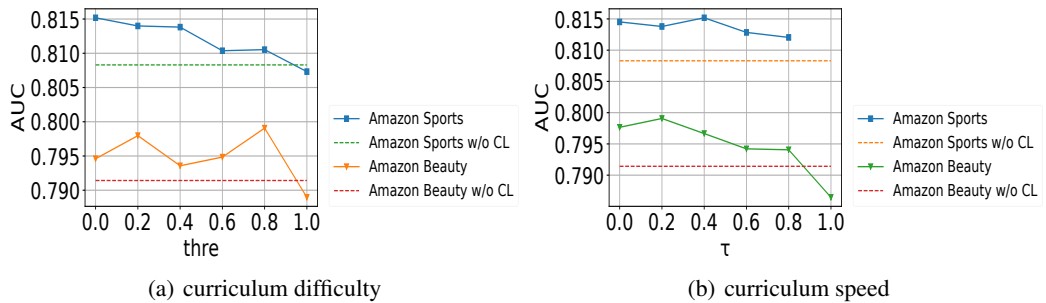

(a) curriculum difficulty   (b) curriculum speed

Figure 2: Ablation on Curriculum

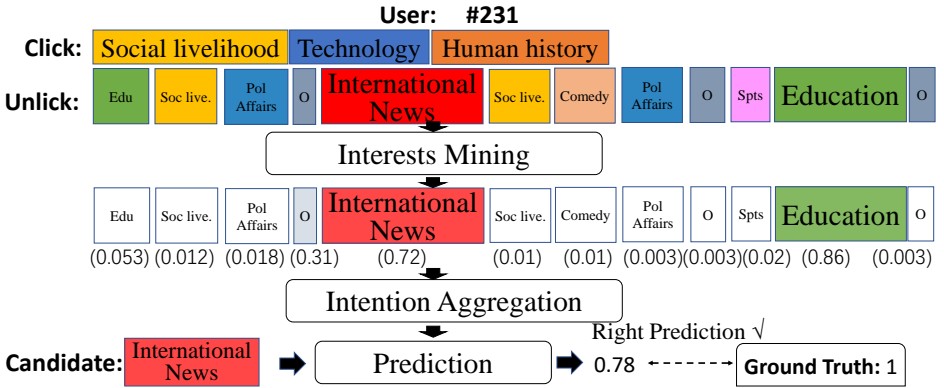

Figure 3: Interests mining from unclick feedback

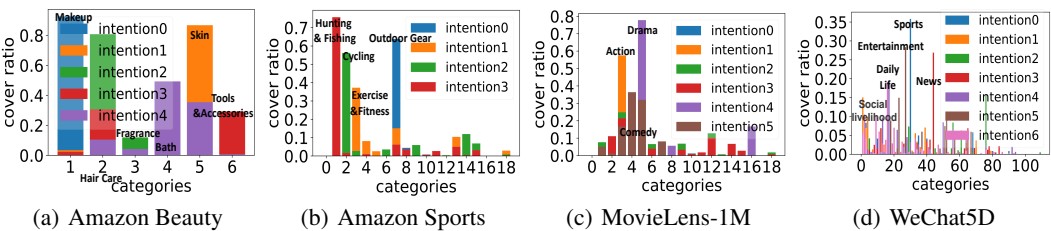

(a) Amazon Beauty (b) Amazon Sports (c) MovieLens-1M (d) WeChat5D

Figure 4: Disentangled Intentions

## 5 Conclusion

In this paper, we propose to learn disentangled representations for user's intentions from multi-feedback. The proposed routing mechanism models the complex relations among multi-feedback and various user intentions, and tackles the impact of noise brought by the multi-feedback. Experimental results show that the learned disentangled representations from multi-feedback could capture comprehensive user intentions and consequently improve the recommendation performance. Moreover, the proposed curriculum further alleviates the impact of data with improper labels in different datasets by adjustable hyperparameters, which can serve as an efficient plug-in for recommendation models. Future work may include explicitly locating the noisy sample labels in multi-feedback data.

## Acknowledgement

This work is supported by the National Key Research and Development Program of China No. 2020AAA0106300 and National Natural Science Foundation of China (No. 62050110, No. 62102222).

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
