# Curriculum Disentangled Recommendation with Noisy Multi-feedback

**Hong Chen**[1]*, **Yudong Chen**[1,2]*, **Xin Wang**[1]†,
**Ruobing Xie**[2], **Rui Wang**[2], **Feng Xia**[2], **Wenwu Zhu**[1]†
[1]Tsinghua University, [2]WeChat Search Application Department, Tencent.
{h-chen20,cyd18}@mails.tsinghua.edu.cn, {xin_wang,wwzhu}@tsinghua.edu.cn,
{ruobingxie,rysanwang,xiafengxia}@tencent.com

## A   Appendix

### A.1   Experimental Details

Table 1: Dataset statistics

|              | Amazon-Beauty | Amazon-Sports | MovieLens-1M | Wechat5D |
|--------------|---------------|---------------|--------------|----------|
| # of users   | 22,342        | 35,590        | 6,039        | 13,340   |
| # of items   | 12,099        | 18,356        | 3,628        | 112,859  |
| # of click   | 176,520       | 277,088       | 836,478      | 749,138  |
| # of unclick | 788,008       | 1,179,266     | 2,138,040    | 7,766,013 |
| # of dislike | 21,847        | 19,203        | 163,515      | 295,504  |

**Datasets**  The dataset statistics are shown in Table 1. Based on the two Amazon datasets and the MovieLens-1M dataset, for each piece of user like or dislike interaction, we generate four pieces of unclick interactions for this user. They include three items that are sampled from the top popular (top 3000 for Amazon datasets and top 1/2 for MovieLens-1M) items at that time and one item randomly sampled from all the items. The whole dataset is chronologically divided to the train, valid, and test dataset by the ratio of 8:1:1. Note that our training and testing phase follow the sequential recommendation setting. For example, if one user's historical behavior is a sequence $\{1, 2, 3, \cdots, 18, 19, 20\}$. Then, we will generate the validation and test samples as follows: two validation samples $\{\left[1, 2, 3, 4, \cdots, 16\right], \left[17\right]\}, \{\left[1, 2, 3, 4, \cdots, 16, 17\right], \left[18\right]\}$ and two test samples $\{\left[1, 2, 3, 4, \cdots, 16, 17, 18\right], \left[19\right]\}, \{\left[1, 2, 3, 4, \cdots, 16, 17, 18, 19\right], \left[20\right]\}$, where the first term in [ ] represents the historical information we use for prediction and the second term is the next item for prediction. We always use all the user's real historical behaviors as the sequential input to the models to predict the next item the user will click.

**Infrastructure**  We implement our model with Tensorflow, and our experiment environments are as follows:

- CPU:Intel(R) Xeon(R) CPU E5-2699 v4 @ 2.20GHz
- RAM: 1TB DDR4
- GPU: 8x GeForce GTX 1080 Ti
- Operating System: Ubuntu 18.04.1 LTS
- Tools: Python2.7, Tensorflow1.13.1, scikit-learn 0.20.3

---

*Equal contributions
†Corresponding Authors.

35th Conference on Neural Information Processing Systems (NeurIPS 2021).

**Hyper-parameter search** We obtain our final parameters with random grid search and the configuration is as follows:

- The number of latent intentions $K \in \{1, 2, \cdots, 8\}$.
- The prior confidence for the unclicked data $\lambda \in \{0.1, 0.2, \cdots, 1.0\}$.
- The learning rate $\in \{0.0001, 0.001, 0.01, 0.1, 1.0\}$.
- The hidden size of each field of feature $\in \{32, 64, 128, 256\}$

In our experiments, we observe that larger $\lambda$ usually brings better performance, which further indicates that the filtering mechanism could locate the real interests from the unclicked data for users considering that if the unclicked data are still noisy, using them to aggregate users' intentions with high confidence will probably harm the recommendation performance.

**Some other ablations** To validate the effectiveness of different parts of our design as one of the reviewers suggest, we conduct ablation studies to show the methods we adopt are effective.

- complete: This variant is our proposed complete method.
- w/o time: This variant is our proposed method without the time-factor attention.
- w/o candidate: This variant is our method without considering the candidate item influence.
- w/o dis loss: This variant is our method by removing the disentangled regularizers in the optimal objective.
- K=1: This variant is our method by fixing the number of disentangled intentions to 1.
- cos sim: This variant replaces the similarity function in our paper with the cosine similarity instead of the original Layer Normalization and inner product.
- w/o residual: This variant removes the residual connection design in the Eq.(1) and Eq.(2) in the paper.

Table 2: Effectiveness of time and candidate factor

| Dataset | Method | | | | | | |
| --- | --- | --- | --- | --- | --- | --- | --- |
| | **complete** | w/o time | w/o candidate | w/o dis loss | K=1 | cos sim | w/o residual |
| Amazon-Sports | **0.8152** | 0.8100 | 0.8036 | 0.8045 | 0.8013 | 0.7782 | 0.8047 |
| Amazon-Beauty | **0.7991** | 0.7802 | 0.7847 | 0.7887 | 0.7814 | 0.7699 | 0.7890 |