# OpenReview forum: "Curriculum Disentangled Recommendation with Noisy Multi-feedback"
_NeurIPS.cc/2021/Conference — NeurIPS 2021 Poster_

### Official Review · Reviewer_foyQ · 2021-07-16

**Rating:** 5
**Confidence:** 4

**Summary:**

The paper introduces an approach to take into account different types of positive and negative feedback in a recommendation model, while accounting for the possible noise contained in the feedback. The proposed model relies in particular on three types of feedback (click, lack of click and dislike). The sequences of different types of feedback are first encoded with a Transformer and then aggregated through a series of MLPs and attention mechanisms into vectors supposed to capture and disentangle the intents of the user.

Independently from this module, the paper also proposes a curriculum learning approach to train the model and allocate with flexibility more weight on either the easier samples or the harder samples at the beginning of the training.

The proposed approach is evaluated on 3 public datasets and 1 proprietary dataset, and compared against several baselines.

**Limitations And Societal Impact:**

The paper did not discuss the limitations of this work and its potential negative societal impact.

A limitation of the work is in the type of feedback considered. The approach is based on "click", "lack of click" and "dislike" signals. However a platform could have access to other types of signals ("like", "add to basket", "add to wishlist", "remove from wishlist", etc.) and it is unclear if the proposed approach could directly use this feedback.

Additionally, the approach seems to suppose that user/item features are discrete and did not cover how continuous features could be handled.

**Main Review:**

### Strengths

- The proposed curriculum approach seems easy to use and general enough to be applied to a wide range of recommendation approaches.

### Weaknesses

- Many choices in the modeling of the "co-filtering dynamic routing" are overly complex, unclear or lack motivations/justifications (see Section "Model design").
- The results of the paper are not reproducible as the code is proprietary.
- The experiments miss the comparison against appropriate and recent baselines (see Section "Empirical methodology"), and did not report statistical significance or error bars.
- Given the high complexity of the approach and its large number of parameters, comparing the runtime of the approach against that of baselines would have been needed.

### Model design

- Based on Section 3.1, the similarity function seems to systematically apply LayerNorm before computing the dot product and softmax operation. What is the motivation for using LayerNorm here?
- In Section 3.2.1, the input item features are projected into an embedding space. It seems that features are assumed to be discrete. Is the approach still adapted for continuous features?
- In Section 3.2.1, it is unclear why the sequence of click interactions, the sequence of "unclick" interactions, and the sequence of dislike interactions are all three considered independently. These sequences might have be linked together. For example, the user may first click on an item A, then check a similar item B and dislike it. One could have simply considered all the interactions in sequence, and after that extract the item representations for each interaction type. Additionally, it is not fully clear if the different Transformer encoders (C-Encoder, U-Encoder and D-Encoder) share some parameters or are totally independent.
- What is the motivation for using a residual connection in Equations (1), (2) and (4)? No justification is provided on that, and the experiments do not report results with/without these to show the need for this.
- The scalars d_ci^(v) and d_uj^(v) in Equation (1) represent the contribution of clicked and unclicked items to the user intention. d_ci^(v) is given by a softmax and is thus strictly positive, however d_uj^(v) is based on the output of an MLP and it is therefore unclear if this quantity is positive or not.
- The MLP used in the definition of d_uj^(v) takes as input the concatenation of F^(v), which contains the base embeddings of all g user features, and z_uj^(v) which was output by the U-Encoder. How is the concatenation performed here given that there are g+1 vectors? Do these vectors have the same dimensions, given that some are base embeddings and some are output by the Transformer encoder?
- In Equation (2), it is unclear what the terms time_ci^(v) and time'_cm^(v) are trying to capture. Given that z_ci^(v) and z_cm^(v) are the output of the Transformer encoder, they should already capture temporality/sequentiality through position embeddings. What is the need then to add a fully-connected layer that takes as input the embedding output by the Transformer *and* the positional embedding? This seems redundant and unnecessary.
- In Equation (4), what are zl_ci^(v) and zl_uj^(v) intended to capture? It is unclear why it is needed to aggregate z_ci^(v) and n'_c^(v) rather than directly using z_ci^(v) in the definition of intent_k^(v).
- The quantity inten_k^(v) in Equation (4) is supposed to capture the interest of user v according to the k-th intent. However inten_k^(v) depends through f_ci^(v) and f_uj^(v) on the candidate item considered. It is unclear why the intention of the user would depend on a candidate item, which should be totally independent from the user as it can be any item.
- The multihead attention introduced in Equation (5) lacks motivations. What is the expected role of the different heads in this context?
- If one omits the bias beta_k in Equation 4, which has only a marginal role, then the sum over the different intents in Equation 6 will actually result in ignoring c_ik^(v) and u_jk^(v), and the notion of intents altogether. Indeed, zl_ci^(v) does not depend on k and sum_k c_ik^(v) = 1. A similar observation holds for zl_uj^(v) and sum_k u_jk^(v). Then the sum over intents in Equation 6 removes all dependencies with respect to an intent: sum_k intent_k^(v) = sum_i d_ci^(v) * f_ci^(v) * zl_ci^(v) + lambda * sum_j d_uj^(v) * f_uj^(v) * zl_uj^(v). Therefore, the prediction made by the model mostly ignores intents, despite intent disentanglement being presented as a main feature of the approach.
- In Equation 6, the dot product term is positive but it is unclear if the output of the MLP term is positive or not.

### Empirical methodology

The sequential recommendation models used as baselines (DIN, SASRec, DFN, SDR) were designed for next-item recommendation: based on a past sequence of interactions, predict the next item. However the experimental setup corresponds to a top-k recommendation scenario: given the past history of interactions, predict the set of test items.

The sequential baselines are then not very appropriate for a task where the test set contains more than a positive item per user, because the different items in the test set won't be all correspond to the next item following the training sequence. Therefore, one needed to either consider a next-item recommendation setup with only one test item to predict, or compare instead against other top-k recommendation baselines.

The authors could have considered the state-of-the-art recommendation model LightGCN [1] as a potential, more recent baseline.

[1] Xiangnan He, Kuan Deng, Xiang Wang, Yan Li, Yong-Dong Zhang, Meng Wang: LightGCN: Simplifying and Powering Graph Convolution Network for Recommendation. SIGIR 2020: 639-648

### Clarity

- It was not fully clear at the beginning of the paper what the "unclick" interaction means. An unclick interaction seems to suggest the user had first clicked on an item and then removed the click. But in fact in this paper an unclick interaction simply corresponds to a lack of click interaction. This should be made more clear and unclick interaction should instead be replaced with lack of click.
- The choice to denote the positive interaction as "click" and the negative interaction has "dislike" creates an asymmetry which can be confusing for the reader. One could question one does not consider the positive interaction "like" which is the positive counterpart of "dislike" and exhibits more confidence than a click, which can be more noisy in its positiveness.
- In Section 4.3, it is not fully clear what the variant c&un of the proposed model consists in.

### Update after author response

I thank the authors for their detailed response to my comments and questions. The extensive efforts that were put in the response is commendable and appreciated. However, there are still several points which, despite the authors' response, remain problematic for me (detailed below), and I will thus keep my initial score.
- **Efficiency & Complexity.** The runtime reported by the authors for the proposed model indeed seems reasonable in comparison to that of DFN. However, the complexity indicated in the response does not seem correct as the reported Transformer complexity ignores the dependence on the length of interaction sequences.
- **Empirical methodology.** The response clarified that the experiments follow a next-item recommendation setting. This will need to be explained in the experimental setup section as in its current version the paper and the supplementary material lack too many details about the evaluation procedure. It is also still not clear whether at inference time, all items are considered as candidates or if a sampling is performed (which should be avoided [2]). With respect to the argument that LightGCN cannot be used as a baseline because of the next-item recommendation setting, I do not agree with the authors. Some of the baselines considered in the paper, such as DeepFM and AutoInt, were not initially designed either for the next-item recommendation task. Therefore, LightGCN could be adapted to the task in the same way as it is done for other baselines. This can be simply done by retraining the model each time new samples are added to the training set.
- **Choice of different behaviors.** The authors claim that "like" behavior is much more rare than "dislike" behavior, however I did not see justifications or references to explain why there would be such an asymmetry.
- **Motivations for using LayerNorm.** The authors mention that using a simple dot product as opposed to a cosine similarity leads to mode collapse, and thus normalization is needed. Then, why not use a simple cosine similarity, by dividing the dot product by the vector norms, instead of using LayerNorm? This argument did not convince me that LayerNorm is necessary or needed in the proposed model. This design choice seems to be guided by the fact that Transformer architectures adopt this, without evidence that this is needed in the proposed model (which vastly differs from a vanilla Transformer). For this choice and others (e.g., use of residual connections) I would have liked to see ablation studies to validate their necessity.
- **General user intentions should be independent of the candidate item.** I still find problematic and unintuitive that user intentions depend on the candidate item. The user-item matching should intuitively be done by comparing a user representation (independent of the specific candidate item) and an item representation (independent of the specific user). Otherwise, the matching process boils down to matching an item with itself, which makes little sense. Having a user representation which depends on the candidate item also seems to lead to inefficient and non-scalable inference, as in the inference phase all items in the datasets need to be considered as candidate items for each user.
- **About the disentangled intentions.** The authors describe beta_k as the center under the k-th intention, and zl_ci and zl_uj as offsets to these intention centers. However, zl_ci and zl_uj are not specific to any intention as they do not depend on intent k; it is then unclear how could they be at the same time the offset from all intent centers beta_k. Moreover, the authors' response did not address the fact that the influence of intents on Eq. 6, with or without beta_k, remain very minor as detailed in my original comment.

[2] Krichene, W., & Rendle, S. (2020). On Sampled Metrics for Item Recommendation. Proceedings of the ACM SIGKDD International Conference on Knowledge Discovery and Data Mining, 1748–1757.

### Update 2

I thank the authors again for the additional clarifications provided in the second round of discussion. In light of the satisfactory response brought by the authors, and although I still consider that some parts of the proposed approach are slightly arbitrary and that significant work will be needed to polish the paper based on our discussion, I will raise my score.

**Time Spent Reviewing:**

6

---

> ### Author Response · Authors · 2021-08-10
> **Reply to the Reviewer(PART II)**
>
> # Empirical Methodology
> The reviewer may have misunderstood our experimental setup. We clarify that all our experiments follow the next-item recommendation setup instead of the top-k recommendation scenario. We use an example to show our next-item recommendation setup. If one user's historical behavior is a sequence $\\{1,2,3,\cdots,18,19,20\\}$. Then, we will generate the validation and test samples as follows: two validation samples $\\{ \Big[ 1,2,3,4,\cdots,16 \Big], \Big[17\Big]\\}$,
> $\\{\Big[1,2,3,4,\cdots,16,17\Big], \Big[18\Big]\\}$ and two test samples $\\{\Big[1,2,3,4,\cdots,16,17,18\Big],\Big[19\Big]\\}$, $\\{\Big[1,2,3,4,\cdots,16,17,18,19\Big],\Big[20\Big]\\}$, where the first term in [ ] represents the historical information we use for prediction and the second term is the next item for prediction.
> We always use all the user's real historical behaviors as the sequential input to the models to predict the next item the user will click.
> Therefore, we conclude that all our experiments follow the next-item recommendation setup, and this setup conforms to the real recommender systems and is widely used in the literature[1,5,6].
>
> Addtionally, LightGCN is not a suitable baseline for our experiments. The reasons are as follows. First, our setting is the next-item prediction but LightGCN is more suitable for top-k recommendation as the reviewer points out. Second, LightGCN is first not closely related to our model, neither related to disentangled recommendation nor related to multi-feedback. Therefore, we did not include it as our baseline.
>
> Finally, our baselines are suitable for our setup, they include SOTA baselines (published in 2020) on disentangled recommendation (SDR[7]) and multi-feedback recommendation (DFN[1]), which are highly-related and recent baselines.
>
> # Clarity
> + Meaning of "unclick": We define items exposed to but not clicked by the user as "unclick" as described in line 103 of the paper, which follows the name of DFN[1]. In our opinions, it's also reasonable to use the name "lack of click" to describe the situation that the items appear in the recommendation column but the user did not click them. It's better to define this kind of behavior at the beginning of the paper as the reviewer suggests and we will modify it in the future version.
> + Choice of different behaviors: Including "Like" behavior will make the model design more symmetric as the reviewer suggests. However, "Like" behavior in real scenario is usually extremely sparse for users. In our model design, if we consider the "like" behavior, we should include one more Transformer Encoder. However, the sparsity of the data will easily make this new Encoder overfit which may in turn deteriorate the model performance. Therefore, we did not include the "Like" behavior in our proposed model.
> + Meaning of the variant c\&un: As mentioned in line 264-265, c\&un corresponds to the situation that we directly use the output of the C-Encoder and U-Encoder to conduct the intention aggregation.
>
> # Limitations and Social Impact
> + Extensions of our method: All the users' behavior could be divided into four categories: explicit/implicit positive and explicit/implicit negative feedback [1] and the explicit positive feedback is too sparse in many scenarios. Therefore, we mainly consider the implicit positive (e.g., click), explicit negative (e.g., dislike) and implicit negative behavior (e.g., unclick). Other behaviors belonging to these categories can be used as the corresponding input in our framework. For example, "remove from wishlist" can be categorized to implicit negative. As for the explicit positive feedback, it's easy to directly add one more transformer encoder followed by the time and candidate factor attention in the Interest Mining step in Figure 1 in our paper.
> + How to handle continuous feature: (see the Model Design section)
>
> [1] Ruobing Xie, Cheng Ling, Yalong Wang, Rui Wang, Feng Xia, and Leyu Lin. Deep feedback network for
> 350 recommendation. Proceedings of IJCAI-PRICAI, 2020.
>
> [2] Jianxin Ma, Chang Zhou, Peng Cui, Hongxia Yang, and Wenwu Zhu. Learning disentangled representations for recommendation. In Advances in neural information processing systems, pages 5711–5722, 2019.
>
> [3] Weiping Song, Chence Shi, Zhiping Xiao, Zhijian Duan, Yewen Xu, Ming Zhang, and Jian Tang. Autoint:Automatic feature interaction learning via self-attentive neural networks. In Proceedings of the 28th ACM International Conference on Information and Knowledge Management, pages 1161–1170, 2019.
>
> [4] Huifeng Guo, Ruiming Tang, Yunming Ye, Zhenguo Li, and Xiuqiang He. Deepfm: a factorization-machine based neural network for ctr prediction. arXiv preprint arXiv:1703.04247, 2017.
>
> [5] Wang-Cheng Kang and Julian J. McAuley. Self-attentive sequential recommendation. In IEEE International Conference on Data Mining, ICDM 2018, Singapore, November 17-20, 2018, pages 197–206. IEEE Computer Society, 2018.
>
> [6] Feng Y, Lv F, Shen W, et al. Deep session interest network for click-through rate prediction[J]. arXiv preprint arXiv:1905.06482, 2019.
>
> [7] Jianxin Ma, Chang Zhou, Hongxia Yang, Peng Cui, Xin Wang, and Wenwu Zhu. Disentangled self supervision in sequential recommenders. In KDD20, August 23-27, 2020, pages 483–491. ACM, 2020.

---

> ### Author Response · Authors · 2021-08-10
> **Reply to the Reviewer(PART I)**
>
> Thanks for the reviewer's careful reading and valuable comments. We address each of the comments from the reviewer one by one as follows.
> # Weaknesses
> + Motivation for Model Design: We clarify the motivation of each part of our model in section Model Design.
> + Code Reproducibility: We will release our codes (including the experiments on the public datasets) when our paper is published.
> + Problems about experimental setup and baselines: The reviewer may have misunderstood our experimental setup and more details are discussed in the Empirical Methodology section.
> + Complexity of our approach: With the following denotations: The dimension for the item embedding: d, the number of disentangled intentions: K, the number of heads in transformer: M. The parameters used in our model design mainly consist of the following parts: 1)Four transformer encoders: $O(4Md^2)$, 2)time,candidate,negative reweighting: $O(6d^2)$,3)the filters in aggregation: $O(4Kd^2)$. So all the parameters we use for the model design are $O((M+K)d^2)$. Since in our experiments we set $M=4$ and $K=\{1,2,\cdots,8\}$, the number of parameters is not large.
> Additionally, we compare the running time of our model with that of DFN[1], an efficient online recommendation algorithm,  by running the same training epochs on Amazon datasets. The results in Table 1 show that the time complexity of our method is acceptable.
> |time(min)|  Method| |
> |----|----|----|
> |Dataset| DFN | ours|
> |Amazon-Beauty| 10.6| 17.4|
> |Amazon-Sports|15.7|26.4|
>
> # Model Design
> + Motivations for using LayerNorm: First, in the process of disentangling user intentions, using cosine similarity is much less vulnerable than dot product when it comes to mode collapse [2], so we need to normalize each vector before the inner product. Second, the whole disentangling process could be regarded as an EM algorithm, and Layer Normalization could boost its convergence. Out of the two considerations, we adopt Layer Normalization for each vector before calculating the similarity.
> + Adaptation for continuous features: We set a basis vector $b_i$ for each category of continuous feature, and the embedding for a continuous feature $m$ could be $m*b_i$. This technique is widely used in literature [3][4].
> + Independence of three encoders: The parameters of the three encoders are independent. As the reviewer suggests, linking the three kinds of behavior could better capture the evolution of the user intentions. However, due to the noise in the historical behavior, considering them together will probably confuse the model according to the results of our pre-experiments. This could be an interesting direction for future work.
> + Motivation for the residual connection: First, we take advantage of the residual connection to learn some additional information. For example, in line 147-148, we use the residual connection to fit the situation when an item is similar to the negative intention but could still be the user's positive intention. The residual part could serve as a subtle modification to our original design. Second, compared to using dense connection, residual connection is easier to optimize.
> + Whether $d_{uj}^{(v)}$ is positive: $d_{uj}^{(v)}$ is positive because in the 4th line of Eq. (1) we use the sigmoid function to resize it to (0,1).
> + Dimension problem: Assume that each user has $g$ fields of features and each item has $r$ fields of features as described in line 178. The dimension for $z_{uj}^{(v)}$ is $d$ that corresponds to $r$ fields, and the dimension for the concatenated feature is $d(r+g)/r$.
> + Motivation for Eq. (2): $time_{ci}^{(v)}$ and $time_{cm}^{(v)\prime}$ try to decide the importance of each historical behavior from the time perspective. Although the output of the Transformer encoder has considered the positional embedding, it cannot tell us which historical behavior has the greatest influence on the user's current intentions. We use this kind of design to explicitly guide the model to pay emphasis on the time serial information with human priors.  This operation is not redundant from the results in Table 2.
> + Motivation for Eq. (4): We obtain $zl_{ci}^{(v)}$ and $zl_{uj}^{(v)}$ with the residual structure for further filtering as mentioned in line 167. The strategy in Eq. (1) will decrease the weight of the potential noisy user historical behavior, and this residual structure is designed to further modify each historical behavior in feature level through the designed sufficient feature interaction with the highly confident negative intention.
> + Consideration for $f_{ci}^{(v)}$ and $f_{uj}^{(v)}$: General user intentions should be independent of the candidate item. However, our final objective is to find whether the user likes the candidate item. The candidate factor design makes the learned intentions more focused on the candidate item. We learn candidate item aware intentions aiming at our final objective and improving the final recommendation performance as shown in Table 2.
> + Motivation for multi-head attention in Eq. (5): We use the multi-head attention to learn different-order combinatorial features. This idea is inspired by AutoInt[2].
> + About the disentangled intentions: It is not appropriate to omit the bias $\beta_k$. $\beta_k$ plays an important role in the method and the reasons are as follows.  When we aggregate the user's intention, $\beta_k$ is used as the center under the $k^{th}$ disentangled intention shared by all users. Then, the term $z1_{ci}$ and $z1_{uj}$ offers information about the user's specific preference information under each category, which can be regarded as the offset from the center. Therefore, the disentangled intentions capture both general characteristics and the user-specific features under the latent category. Ignoring $\beta_k$ is not proper.
> + About Eq. (6): The output for Eq. (6) need not to be guaranteed to be positive or negative because before calculating the loss, we will send it to a sigmoid layer in Eq. (7), whose output is always positive.
> |     |       |Method|  |
> |  ----  | ----  |----| ----|
> | Dataset | complete | w/o time | w/o candidate|
> | Amazon-Beauty | 0.8152| 0.8100| 0.8036 |
> | Amazon-Sports  |0.7991| 0.7802| 0.7847|

---

> ### Author Response · Authors · 2021-08-16
> **Reply to the reviewer's updated comments**
>
> We appreciate reviewer’s prompt update upon our response. Please see our response to the reviewer’s update as follows.
>
> + Efficiency & Complexity:
> We apologize that we misunderstood your comments in the first response. In our first reply, the complexity $O((M+K)d^2)$ we report is from the aspect of the number of parameters. In this response, the calculation time complexity for one training sample is O($MH^2d^2+KHd^2$), where H is the length of the historical sequence.
>
> + Empirical methodology:
>     1) We thank reviewer’s suggestions and will certainly add more detailed explanations in the experimental setup in our future version for more clarity.
>     2) In our experiments, all items are considered as candidates (i.e., no sampling is performed).
>     3) In terms of the LightGCN baseline, we have followed the reviewer’s suggestion and conduct an additional experiment to compare our proposed method with LightGCN in the following table, which shows LightGCN fails to outperform our proposed approach.
> |     |       |Method|
> |  ----  | ----  | ---- |
> | Dataset | ours| LightGCN|
> | Amazon-Beauty | 0.8152| 0.7715|
> | Amazon-Sports  |0.7991|0.7587|
>
>
> + Choice of different behaviors:
> We follow the reviewer’s suggestion to list the ratio of each kind of feedbacks over all the feedbacks in the Wechat5D dataset, which demonstrates the sparsity of “like” behavior and the asymmetry in real-world dataset. Actually, for the choice of different behaviors (feedbacks), we follow the practice in previous work, DFN [1].
> |  like   |  dislike    | click| unclick|
> |  ----  | ---- | ---- | ---- |
> |  0.04810%  |3.352%  |8.499% |88.10% |
>
>
> + Motivations for using LayerNorm:
> As the reviewer suggests, we conduct ablation studies as follows:
> 	1. cosine similarity: We conduct experiments with cosine similarity. Specifically, we replace the layer normalization with L2 normalization which is used to calculate the cosine similarity. The experimental results show that cosine similarity does not perform as good as LayerNorm (see Table below).
> 	2. Others:
>
>         a) w/o Residual connection: We remove the residual connection in Eq. (1) and Eq. (2). The experimental results obtained without the residual connection are not as good as those with the residual connection.
>
>          b)w/o Filtering with negative intentions in Eq. (4): We remove the negative feature filtering in Eq. (4). The experimental results show that the negative feature filtering benefits the intention learning.
> |     |Amazon-Sports|Amazon-Beauty |
> |  ----  | ----  |----|
> | cosine similarity| 0.7782| 0.7699|
> | w/o residual connection|0.8047|0.7890|
> | w/o filtering with negative intention | 0.7918 | 0.7653|
> | ours| 0.8152| 0.7991|
>
>
> + Intentions and candidate item:
> We conduct experiments where the user intention do not depend on the candidate item. In the experiments, we find that making user intention dependent on the candidate item can outperform the setting where user intentions do not depend on the candidate item (see table below).
> The experimental results agree with our proposed method, which has a design of the “candidate item aware user intention learning”. This design is also inspired by the previous works[2,3,4], indicating that taking candidate item into consideration will make the learned user representation more expressive.
> We agree that this candidate item aware design may also result in the tradeoff between accuracy and efficiency as the reviewer suggests.
> To our final version, we will include both of the two designs.
> |     |       |Method|
> |  ----  | ----  |----|
> | Dataset | ours| w/o candidate|
> | Amazon-Beauty | 0.8152| 0.8036 |
> | Amazon-Sports  |0.7991| 0.7847|
>
> + About disentangled intentions:
> 1.	Relations between $zl_{ci}^{(v)}$, $zl_{uj}^{(v)}$ and intent k:
> As shown in the third line in Eq. (4), the coefficient $d_{ci}^{(v)}*f_{ci}^{(v)}*c_{ik}^{(v)}$ is proportional to the probability of $z1_{ci}^{(v)}$ belonging to the $k^{th}$ latent intention of user v, we aggregate all the items clicked by the user weighted by probability, which is the same as the process to approximate the center in the gaussian EM algorithm. Additionally, instead of using $z1_{ci}^{(v)}$ only, we use ($z1_{ci}^{(v)}+\beta_k$) to aggregate the user’s intents under the $k^{th}$ latent category. $\beta_k$ is the center which reflects the general preference under the $k^{th}$ latent category, and $z1_{ci}^{(v)}$ indicates the user’s specific preference which serves as the “offset” from the center.
> 2.The influence of the intents on Eq. (6) from both the mathematical perspective and empirical perspective:
>     + From the mathematical perspective, if the user has higher intention under the $k^{th}$ latent category, the final coefficient for $\beta_k$ (equal to $\sum_{i=1}^{m}d_{(ci)}^{(v)}f_{ci}^{(v)}c_{ik}^{(v)}$) will be larger based on the following equation Eq. (I) , and the $z1_{ci}^{(v)}$ term serves as the user specific interests term. Therefore, the results in Eq. (6) still relies on each intent k considering the term $\sum_{k=1}^{m}\sum_{i=1}^{m} c_{ik}^{(v)}\beta_k$ in the last line of the following equation Eq. (I). Since $c_{ik}^{(v)}\beta_k$ has two terms related to k, we cannot merge them using $\sum_{i=1}^{k}c_{ik}^{(v)}=1$. Instead, we should calculate the coefficients before $\beta_k$ as $\sum_{i=1}^{m}d_{(ci)}^{(v)}f_{ci}^{(v)}c_{ik}^{(v)}$.
>
>         $\sum_{k=1}^K<inten_k^{(v)},h_t^{(v)}>$
> $= \sum_{k=1}^K< \sum_{i=1}^{m} d_{ci}^{(v)} \cdot f_{ci}^{(v)} \cdot c_{ik}^{(v)} (z1_{ci}^{(v)} +\beta_k)  + \lambda
>     \sum_{j=1}^{l} d_{uj}^{(v)} \cdot
>     f_{uj}^{(v)} \cdot u_{jk}^{(v)} \cdot (\textbf{z1}_{uj}^{(v)}+\beta_k),h_t^{(v)}> $
> $=<\sum_{i=1}^md_{ci}^{(v)} \cdot f_{ci}^{(v)}(\textbf{z1}_{ci}^{(v)}+\sum_{k=1}^Kc_{ik}^{(v)}\beta_k)+ \lambda \sum_{j=1}^{l} d_{uj}^{(v)} \cdot
> f_{uj}^{(v)}(z1_{uj}^{(v)}+\sum_{k=1}^Ku_{jk}^{(v)}\beta_k ),h_t^{(v)} >  (I) $
>
>         The above statement indicates that the disentangled intents do have influence on Eq. (6).
>
>     + From the empirical perspective, the disentangled intent learning plays the role as a bridge that connects the user past behaviors and future behaviors. We first learn the disentangled intents from the user’s multi-feedbacks and then make predictions based on all the disentangled intents. Therefore, the disentangled intents in our design help to better capture the user’s preferences over different aspects.
> To validate the above claim, we conduct an additional ablation studies to show the effectiveness of the disentangled intents in the following table. We first consider removing the disentangled loss from our regularizer (w/o disentangled loss) and the results show that making the representations disentangled indeed is helpful for the final recommendation. Also, we conduct ablation studies to make the number of latent categories equal to 1, indicating that we learn one unified entangled intent for each user, which reduces to the situation that the prediction does not rely on any intents mentioned by the reviewer. The results also show that one single unified entangled intent is not enough to precisely capture the user’s comprehensive preference, which is also claimed in [5].
> |     |Amazon-Sports|Amazon-Beauty |
> |  ----  | ----  |----|
> | w/o disentangled loss| 0.8045|0.7887|
> |K = 1 | 0.8013 | 0.7814|
> |ours|0.8152|0.7991|
>
>          The above empirical results also demonstrate the disentangled intents learned by our model tend to have influence on the final model prediction accuracy.
>
> [1] Ruobing Xie, Cheng Ling, Yalong Wang, Rui Wang, Feng Xia, and Leyu Lin. Deep feedback network for 350 recommendation. Proceedings of IJCAI-PRICAI, 2020.
>
> [2] Guorui Zhou, Xiaoqiang Zhu, Chenru Song, Ying Fan, Han Zhu, Xiao Ma, Yanghui Yan, Junqi Jin, Han 442 Li, and Kun Gai. Deep interest network for click-through rate prediction. In Proceedings of the 24th ACM 443 SIGKDD International Conference on Knowledge Discovery & Data Mining, pages 1059–1068, 2018.
>
> [3] Feng Y, Lv F, Shen W, et al. Deep session interest network for click-through rate prediction[J]. arXiv preprint arXiv:1905.06482, 2019.
>
> [4] Lyu Z, Dong Y, Huo C, et al. Deep match to rank model for personalized click-through rate prediction[C]//Proceedings of the AAAI Conference on Artificial Intelligence. 2020, 34(01): 156-163.
>
> [5] Xiang Wang, Hongye Jin, An Zhang, Xiangnan He, Tong Xu, and Tat-Seng Chua. Disentangled graph 343 collaborative filtering. In Proceedings of the 43rd International ACM SIGIR Conference on Research and 344 Development in Information Retrieval, pages 1001–1010, 2020

---

> ### Author Response · Authors · 2021-08-18
> **Reply to the Update2**
>
> Thank the reviewer for carefully reading our paper and the efforts put in the discussion, which helps to further improve our paper.

---

### Official Review · Reviewer_jp7S · 2021-07-16

**Rating:** 7
**Confidence:** 4

**Summary:**

The paper proposes a curriculum disentangled recommendation model to learn disentangled representations from multi-feedback data. Specifically, the authors design a co-filtering routing mechanism to capture the relations among multi-feedbacks and denoise the representations at the feature level. Then they utilize the curriculum learning to evaluate sample difficulties to do denoising at the label level. The extensive experiments show the effectiveness of the proposed method on recommendation tasks.

**Limitations And Societal Impact:**

No discussions about the potential negative societal impact.

**Main Review:**

Pros:
-	The proposed method tries to learn the disentangled representation for the recommendation problem from multi-feedback with noisy information. The task is well-motivated and important for real-world scenarios.
-	The paper originally considers two-level noisy information. One is feature-level noise, and another is label-level noise.
-	The paper is well-written and easy to follow.

Cons:
-	The authors should give the definitions of feature level noise and label level noise. The label-level noise is easy to understand. But the feature level noise is confusing. For example, what kind of click action would be regarded as noise? The noisy unclicked items might be useless items that are different from the noise in the labels.
-	The hyper-parameters of curriculum learning need to be searched and predefined manually. It is difficult to find the optimal parameters for different datasets, especially in the online scenario.
-	How to decide the number of k latent intentions is not clear.
-	The code is not provided. It is hard for readers to reproduce the results in the experiments.


**Time Spent Reviewing:**

4

---

> ### Author Response · Authors · 2021-08-10
> **Reply to the Reviewer**
>
> Thanks for the reviewer's careful reading and valuable comments. We address each of the comments from the reviewer one by one as follows.
> + We will add the following definitions in the future version.
>     + Label level noise: Incorrect label for the (user, candidate item) pair.
>     + Feature level noise: Inaccurate learned representations that can not reflect the user's true intentions.
>
>     For example, one young user likes cartoon and game. He first clicked one “game” article. After reading this article, he misclicked one “politics” article next to the “cartoon” article. After that, he clicked the “cartoon” article. This series of action will result in the “game,politics,cartoon” click historical behavior sequence. During training, we will use “game” as the historical behavior to predict the “politics”, which is labeled as positive but in fact caused by misclicking. This results in the label level noise. Also, we will use “game,politics” as historical behavior to predict “cartoon”. We need to aggregate the intention from the two historical behavior, but the latent representation aggregated from the “politics” can not reflect the user's true intention. This results in the feature level noise.
> + Hyper-parameters of curriculum learning need to be searched with random grid search method in our experiments as mentioned in the Supplementary. We fix $\mu$ to 10 to make the model first concentrate enough on particular level of difficulty. We set τ in $\\{0.0,0.1,0.2,\cdots,1.0 \\}$ because it is a decay factor smaller than 1 and we uniformly split the space. We set $thre$ in $\\{0.0,0.2,0.4,\cdots,1.0 \\}$ because the scope of the difficulty is $[0,1]$ and we split the space uniformly.  In online scenario, how to fast adapt these parameters could be an interesting topic for our future work.
> + Decide the number of $K$:  As we mentioned in line 18 in the Supplementary, we search it as a hyper-parameter with the random grid search method. For the range of K, we follow the prior work[1,2] and set it as $\\{1,2,3,\cdots,8\\}$.
> + We will release our codes (including the experiments on the public datasets) when our paper is published.
>
> [1] Jianxin Ma, Chang Zhou, Peng Cui, Hongxia Yang, and Wenwu Zhu. Learning disentangled representations for recommendation. In Advances in neural information processing systems, pages 5711–5722, 2019.
>
> [2] Jianxin Ma, Chang Zhou, Hongxia Yang, Peng Cui, Xin Wang, and Wenwu Zhu. Disentangled self supervision in sequential recommenders. In KDD20, August 23-27, 2020, pages 483–491. ACM, 2020.

---

> > ### Comment · Reviewer_jp7S · 2021-08-28
> > **Discussion**
> >
> > I would like to thank the authors for their response. This has addressed most of the concerns that I raised in my review. I am updating my score to 7.

---

> > > ### Author Response · Authors · 2021-08-28
> > > **Reply to the discussion**
> > >
> > > Thank the reviewer for affirmation to our work and the suggestions that help further to improve our paper.

---

### Official Review · Reviewer_Bacg · 2021-07-19

**Rating:** 8
**Confidence:** 4

**Summary:**

This paper proposes a model to learn disentangled representations for user intentions from complex and noisy multi-feedback. The model can capture complex relations among multi-feedback and various user intentions, and filter out feature-level noise in multi-feedback. Moreover, this paper proposes an adjustable curriculum to alleviate the impact of label-level noise in a more controllable way.



**Limitations And Societal Impact:**

In Section 3.2.1, you take the time and candidate item factors into consideration, how these factors affect the performance of the model? You need to add an experiment to demonstrate it.

In In Algorithm 1, \mu changes as the training goes on. Can $thre$ also change as the training goes on?


**Main Review:**

Originality: This paper proposes a new method to utilize multi-feedback data and a new adjustable curriculum learning method. The contributions are clear and the related work is adequate. The method combines existing components to form a reasonable solution, so the overall technical novelty is not big.

Quality: The proposed method is clear and the experiment is sufficient.

Clarity: This paper is well written and organized. And most of the proposed method and the experiment are clear. But I have doubts on some details. In second row and fourth row of Eq. (1), why does d_{ci}^{v} use softmax and d_{ui}^{v} use sigmoid? And in fourth row of Eq. (1), should the output of MLP between 0 and 1 like sigmoid? In Eq. (4), why are z_{ci}^{v}-n_{c}^{v} and z_{ci}^{v} \odot n_{c}^{v} also inputs to MLP?

Significance: The idea of filtering the noise in both clicked and unclicked items from disliked items is particular. The proposed curriculum may be applied to other tasks. The extensive experiments shows that the model achieve state-of-the-art performance.


**Time Spent Reviewing:**

4 hours

---

> ### Author Response · Authors · 2021-08-09
> **Reply to the Reviewer Comments**
>
> Thanks for the reviewer's careful reading and valuable comments. We address each of the concerns from the reviewer one by one as follows.
> + About Eq.(1): The softmax and sigmoid problem: the softmax and sigmoid function are all used to make all the coefficients positive and between (0,1) for stable training.
>     However, when designing the model, we just consider giving higher weights to the item that could reflect the user's preference but ignore the symmetry of model design.
> + Fourth row of Eq. (1), I guess the reviewer may misunderstand the scope of the sigmoid layer. The sigmoid layer will map the sum of the similarity and the MLP output to (0,1) instead of the first term.
> + About Eq. (4): We hope the high confidence $n_{c}^{(v)}$ could serve as a feature filter for each historical behavior feature, so we design different kinds of interaction for more sufficient feature modification and filtering.
> + We conduct the ablations about the time and candidate factor on the two amazon datasets as shown in Table 1. Since these two are only two small designs in our whole model, we did not show the results in our paper. The results show that both designs are useful for recommendation. The time factor design uses the human prior to explicitly help the model to better capture the user's current intention. The candidate factor design makes the learned intentions more focused on the candidate item, aiming at our final objective (to judge whether the candidate item satisfies the user's intention).
> |     |       |Method|  |
> |  ----  | ----  |----| ----|
> | Dataset | complete | w/o time | w/o candidate|
> | Amazon-Beauty | 0.8152| 0.8100| 0.8036 |
> | Amazon-Sports  |0.7991| 0.7802| 0.7847|
> + $thre$ represents the difficulty of the curriculum. According to the existing curriculum works, the changes of the difficulty may depend on both the model and the datasets. This could be an interesting future work for our algorithm.

---

### Decision · Program_Chairs · 2021-09-27

**Decision:**

Accept (Poster)

**Comment:**

Overall strong scores, and reviewers are mostly aligned in finding the paper to be above the bar. Scores were initially more mixed, though after the discussion the weakest score was improved (to "borderline"). Reviewers praised the originality of the idea, and found the model performance through the experiments convincing enough. Several issues were raised regarding clarity and other specific details, and an in-depth discussion took place to resolve most of these issues.

Ultimately the scores are strong, and the rebuttal was fairly persuasive. The reviewers do highlight several issues (and in spite of the strong scores, the reviews are not particularly gushing in the text). But the issues raised seem mostly in terms of clarity, or otherwise are adequately  addressed during the rebuttal phase.